# Artificial intelligence in sepsis early prediction and diagnosis using unstructured data in healthcare

Kim Huat Goh [1,4 ✉], Le Wang [1,4 ✉], Adrian Yong Kwang Yeow [2], Hermione Poh[3], Ke Li[3], Joannas Jie Lin Yeow [3] & Gamaliel Yu Heng Tan[3]

Sepsis is a leading cause of death in hospitals. Early prediction and diagnosis of sepsis, which is critical in reducing mortality, is challenging as many of its signs and symptoms are similar to other less critical conditions. We develop an artificial intelligence algorithm, SERA algorithm, which uses both structured data and unstructured clinical notes to predict and diagnose sepsis. We test this algorithm with independent, clinical notes and achieve high predictive accuracy 12 hours before the onset of sepsis (AUC 0.94, sensitivity 0.87 and specificity 0.87). We compare the SERA algorithm against physician predictions and show the algorithm's potential to increase the early detection of sepsis by up to 32% and reduce false positives by up to 17%. Mining unstructured clinical notes is shown to improve the algorithm's accuracy compared to using only clinical measures for early warning 12 to 48 hours before the onset of sepsis.

[1] Nanyang Business School, Nanyang Technological University, Singapore, Singapore. [2] School of Business, Singapore University of Social Sciences, Singapore, Singapore. [3] Group Medical Informatics Office, National University Health System, Singapore, Singapore. [4] These authors contributed equally: Kim Huat Goh, Le Wang. ✉email: akhgoh@ntu.edu.sg; lwang033@e.ntu.edu.sg

Sepsis is a leading cause of death in United States hospitals[1], accounting for half of all hospital deaths[2]. Early prediction of sepsis is crucial in preventing mortality, given that sepsis management is highly time sensitive[3]. Based on international medical guidelines[4], early fluid resuscitation is recommended to commence within the first 3 h to stabilize sepsis-induced tissue hypoperfusion[4], and administration of intravenous antimicrobials is recommended to commence at the earliest possible time, specifically within 1 h of sepsis[4–6]. Further, sepsis mortality increases significantly with each hour of delay in antimicrobials administration[5,6]. As sepsis management is often based on a standardized management approach[4], early sepsis identification may be practically challenging and operational constraints in healthcare delivery can lead to unacceptably high mortality rates. For example, delay in communication among clinicians, nurses, and pharmacist exacerbates delay in sepsis management. Hence, the early prediction of sepsis before its onset in a patient gives clinicians additional lead time to plan and execute treatment plans.

Most of the existing methods for sepsis diagnosis and early prediction only take advantage of structured data stored in the electronic medical records (EMR) system[7–9]. However, research has shown that about 80% of the clinical data in EMR systems consist of unstructured data—i.e., data stored without a pre-determined or standardized format[10]. Common examples of such unstructured data are free-form text (e.g., clinical notes) or images (e.g., radiological images). These unstructured clinical data contain rich information, i.e., additional clinical details not captured in the EMR structured data fields. Physicians use such unstructured clinical data fields to record "free-form" clinical notes as structured data is designed to store only pre-determined discrete data (e.g., patient vital signs). Physicians also depend on unstructured data to review judgments and critical clinical information entered by other clinicians to gain a better understanding of a patient's condition or effects of their treatment. Consequently, unstructured EMR data are a potentially rich data source to develop better artificial intelligence (AI) tools, especially for medical conditions such as sepsis, where early symptoms are ambiguous and difficult to recognize. To this end, more recent work has incorporated text-mining of clinical notes to improve the accuracy of early sepsis prediction[11,12].

In the area of clinical notes mining, researchers have used natural language processing (NLP) mainly to identify and extract medical events, medication information, and clinical workflow from unstructured text data stored in EMR systems[13–16]. While most NLP applications on clinical notes mining have mainly focused on identifying and extracting concepts, our study extends prior text-mining sepsis predicting algorithms[11,12] to diagnose and predict sepsis with higher accuracy and with a longer early warning lead time of up to 48 h.

It is a well-known fact that the diagnosis of sepsis is often equivocal due to the varied nature of infection sources and wide-ranging patients' responses[7]. Prior research that used text mining to augment sepsis prediction advocates the use of extracting common words[11,12] that are found in clinical notes to improve model predictive accuracy. In our study, we extend this stream of research by showing that modeling for common topics can lead to higher levels of predictive accuracy and the ability to predict up to 48 h prior to the onset of sepsis. We argue that the use of topics—as opposed to words—is preferred, as lexicographical topics are more stable compared to individual words[17,18]. Further, this method is more generalizable because it can mitigate challenges such as idiosyncratic words used by different physicians due to differences in writing styles.

Here, we developed a topic-based, NLP-enabled AI algorithm that combines the NLP analysis of physicians' clinical notes with structured EMR data to improve our ability to predict the risk of sepsis. Specifically, our algorithm extracts, analyzes, and summarizes physicians' clinical notes and combines these sets of summarized clinical information with structured clinical variables to classify if a patient has sepsis at the time of analysis. For patients who are not classified as having sepsis at the time of analysis, the algorithm will then predict the risk of those patients having sepsis in the following 4, 6, 12, 24, and 48 h. Unlike prior NLP sepsis predictive models, our model operates in situations where the prevalence of sepsis is as low as 6%—equivalent to the prevalence of sepsis in hospitals observed in historical studies[19]. In addition, we found that mining clinical notes provide significant improvements to the predictive accuracy over structural variables for predictions 12–48 h ahead of the onset of sepsis; this is also beyond the accuracy of physician predictions. As discussed above, under standard sepsis management protocol, every hour delay in completion of the administration of the 3-h bundle is found to be associated with a 4% increase in mortality[20]; thus the ability of our algorithm to speed up diagnosis and early detection would potentially reduce overall mortality in hospitals in a significant way.

## Results

**Data and data processing**. This study examines patients admitted to a Singapore government-based hospital (details of the sampling are in "Methods"). We construct the dependent variable for sepsis using the ICD-10 code for sepsis, severe sepsis, or sepsis shock, and the presence of ICU admission. The hospital practice is such that patients diagnosed as having sepsis are transferred to the ICU ward; hence, patients with at least one of these ICD-10 codes and admitted to the ICU ward are allocated to the sepsis case cohort. All other patients not meeting these criteria are allocated to the non-sepsis control cohort. There are 240 sepsis patients in the training and validation sample and 87 sepsis patients in the test sample. We define the sepsis onset time as the ICU ward admission time as per the hospital's practice.

At the patient encounter level, the prevalence of sepsis in our case-control sample is 6.15%, which is equivalent to the estimated sepsis's prevalence of around 6% in hospitals[19]. Given that the data is imbalanced, we applied the synthetic minority over-sampling technique (SMOTE) to achieve 1:1 balanced data for sepsis cases and non-sepsis controls (at the clinical note level). Prior literature argued that oversampling (instead of under-sampling) will result in more accurate models[21–23] and SMOTE has been used in earlier studies that develop machine learning classifiers for other clinical conditions such as oral cancer detection[22] and cell identification/classification[24,25] where the prevalence of the positive cases are low. For comparative purposes, we also develop, test, and report the models without any oversampling to present the possibility of operating this algorithm in a normal clinical environment where the prevalence of sepsis is relatively low.

**Sepsis early risk assessment (SERA) algorithm**. Figure 1 outlines the steps used to develop the SERA algorithm. The unit of analysis for the SERA algorithm is each patient consultation instance given that the algorithm utilizes clinical notes and structured data to make a sepsis risk assessment. We adopt this unit of analysis to ensure that the algorithm can operate in a typical clinical context where clinicians consult, assess, and diagnose patients. SERA algorithm consists of two inter-linked algorithms—a diagnosis algorithm and an early prediction algorithm. The diagnosis algorithm determines if the patient has sepsis at the time of consultation and if not, the early prediction algorithm will

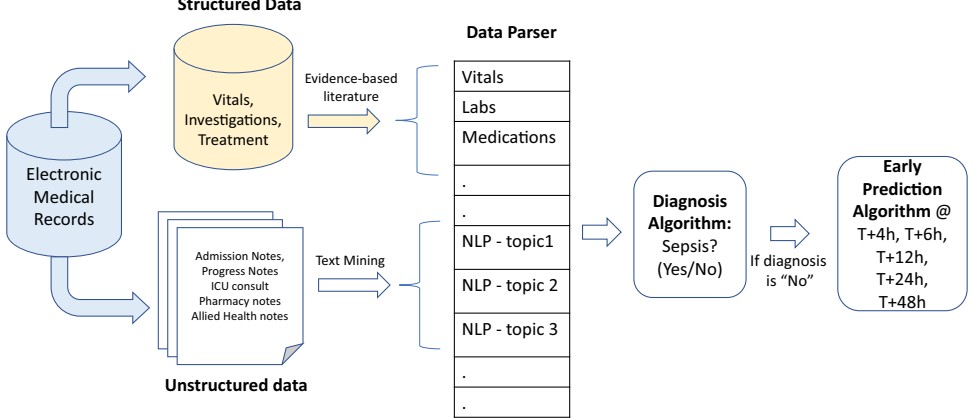

**Fig. 1 Setup of SERA algorithm.** The flow diagram shows the steps used to develop the SERA Algorithm. Both structured data (vitals, investigations, and treatment) and unstructured data (clinical notes) are used in the process of diagnosing and predicting sepsis.

**Table 1 List of structural variables used in the predictive algorithm.**

| Category | Predictors |
|---|---|
| Patient information | Age, gender |
| Vital signs | Blood pressure, heart rate, temperature, oxygen saturation, respiratory rate |
| Investigations | Total white cell, culture results[a], lactate, high-sensitivity C-reactive protein, procalcitonin, arterial blood gas |
| Treatment | Use of vasopressor, use of antibiotics |

[a]The cultures include: the culture of urine, blood culture (anaerobic/aerobic), general culture, wound culture, stool culture, molecular, fluid culture, respiratory culture, cerebrospinal fluid culture, fungus smear, tissue culture, fluid culture (bactec bottle), fluid culture (aerobic), ear culture, genital culture, sterility testing, tip culture, wound culture (aerobic), appearance, stain results, or any real-time polymerase chain reaction. Cultures are typically used to confirm test presence of carbapenem-resistant enterobacteriaceae, methicillin-resistant staphylococcus aureus, clostridium difficile antigen, clostridium difficile toxin, clostridium difficile influenza, influenza B (real-time polymerase chain reaction), influenzae virus, strep pneumoniae antigen, carbapenem-resistant enterobacteriaceae screening, hepatitis viral load, cytomegalovirus, methicillin-resistant staphylococcus aureus screening, legionella antigen, sarcoptes scabiei, herpes simplex virus.

determine the patient's risk of having sepsis in the next 4, 6, 12, 24, and 48 h.

For the diagnosis algorithm, we combined EMR system clinical notes for each consultation with the most recent structured variables available in the EMR system, as listed in Table 1. These data are used to classify if the patient has sepsis at the time of consultation. For the early prediction algorithm, the data structure is similar to that of the diagnosis algorithm except that the early prediction algorithm will not consider any patient consultations from the sample when the patient has been confirmed to have sepsis (i.e., transferred to an ICU). This omission is necessary to prevent positively biasing the predictive power of the algorithm because these positive sepsis consultations would have traits that are strongly associated with sepsis conditions.

**Processing of clinical notes and machine learning**. Before using the unstructured clinical notes as predictors in the algorithm, we applied NLP to these notes, specifically using the latent Dirichlet allocation (LDA) topic modeling algorithm. The major topics found in the progress notes are extracted, numerically weighted, and combined with the structured variable data in Table 1. By applying the NLP LDA model to the clinical notes dataset, we identified 100 common text topics that are in turn classified under one of the following seven categories: (1) clinical status, (2) communication, (3) laboratory tests, (4) non-clinical status, (5) social relationships, (6) symptom, and (7) treatment. The numerical loadings on the 100 topics, together with the structured data, are used as predictors in the diagnosis algorithm and the early prediction algorithm.

In the diagnosis algorithm, the data is subjected to a voting ensemble machine learning algorithm. For comparative purposes,

we also used dagging, and gradient boosted trees (GBT) as two alternative classifiers. For consultations where the patient is classified as not having sepsis, the early prediction algorithm will then predict if the patient has a high risk of having sepsis in the next 4, 6, 12, 24, and 48 h using the voting ensemble machine learning method. The dependent variables for the early prediction algorithm are whether the patient will have sepsis in the next 4, 6, 12, 24, and 48 h, respectively.

The trained and validated models are subsequently tested using the independent, hold-out test sample. Methodological details of the text-mining, diagnosis, and early prediction algorithms are provided in "Methods".

We report the test results of the SERA algorithm for both oversampled (SMOTE) and non-oversampled (non-SMOTE) data. The SMOTE models present the results typically observed in machine learning predictive models where the prevalence of sepsis cases is high and equal to that of the non-sepsis cases[12] (Table 2). The non-SMOTE models present the performance of the model in typical clinical settings where the prevalence of sepsis is low (Table 3). For brevity, we focus on describing the results for the SMOTE models as in prior studies[22,24,25].

For the diagnosis algorithm, our test sample yielded an AUC of 0.94, sensitivity 0.89, specificity 0.85, and positive predictive value (PPV) 0.85. The AUC of the diagnosis algorithm is higher than those of prior studies[7,9,11,12,26–31], which range from 0.64 to 0.92.

The SERA algorithm predicts if a patient has a high risk for sepsis before being diagnosed with sepsis by physicians in the hospital. The algorithm predicting 48 h ahead of the onset of sepsis has an AUC of 0.87. The AUC improves to 0.90, 24 h prior to sepsis, and up to 0.94, 12 h prior to sepsis. Notably, our algorithm's prediction at the 12-h lead time has a higher AUC, sensitivity, specificity, and PPV than prior research[8,11,12,32]. Our 24-h lead time early prediction also has a high AUC of 0.90 and at

**Table 2 Statistics of diagnosis and early prediction algorithm (SMOTE).**

| Diagnosis algorithm | | | | | | | |
|---|---|---|---|---|---|---|---|
| Outcome predict if the patient has sepsis | Voting AUC | Sensitivity | Specificity | PPV | NPV | Dagging AUC | GBT AUC |
| At the present time | 0.94 | 0.89 | 0.87 | 0.85 | 0.90 | 0.92 | 0.94 |
| Early prediction algorithm | | | | | | | |
| Outcome predict if patient will have sepsis | Voting AUC | Sensitivity | Specificity | PPV | NPV | Dagging AUC | GBT AUC |
| 48 h later | 0.87 | 0.78 | 0.77 | 0.77 | 0.78 | 0.83 | 0.83 |
| 24 h later | 0.90 | 0.81 | 0.80 | 0.80 | 0.80 | 0.86 | 0.86 |
| 12 h later | 0.94 | 0.87 | 0.87 | 0.87 | 0.87 | 0.92 | 0.92 |
| 6 h later | 0.92 | 0.88 | 0.81 | 0.82 | 0.87 | 0.90 | 0.93 |
| 4 h later | 0.92 | 0.86 | 0.80 | 0.81 | 0.86 | 0.85 | 0.92 |

SMOTE applied to clinical notes to achieve a balanced sample of sepsis and non-sepsis case entries. SERA algorithm uses the voting algorithm; dagging and GBT algorithms are presented for comparative purposes.

**Table 3 Statistics of diagnosis and early prediction algorithm (in low prevalence condition without SMOTE).**

| Diagnosis algorithm | | | | | | | | |
|---|---|---|---|---|---|---|---|---|
| Outcome predict if the patient has sepsis | Voting Prevalence | AUC | Sensitivity | Specificity | PPV | NPV | Dagging AUC | GBT AUC |
| At the present time | 0.177 | 0.94 | 0.89 | 0.87 | 0.59 | 0.97 | 0.92 | 0.94 |
| Early prediction algorithm | | | | | | | | |
| Outcome predict if patient will have sepsis | Voting Prevalence | AUC | Sensitivity | Specificity | PPV | NPV | Dagging AUC | GBT AUC |
| 48 h later | 0.012 | 0.87 | 0.76 | 0.76 | 0.04 | 0.99 | 0.82 | 0.85 |
| 24 h later | 0.010 | 0.90 | 0.81 | 0.79 | 0.04 | 0.99 | 0.88 | 0.89 |
| 12 h later | 0.008 | 0.94 | 0.88 | 0.82 | 0.04 | 0.99 | 0.92 | 0.93 |
| 6 h later | 0.002 | 0.92 | 0.88 | 0.83 | 0.01 | 0.99 | 0.90 | 0.93 |
| 4 h later | 0.001 | 0.92 | 0.89 | 0.87 | 0.01 | 0.99 | 0.92 | 0.94 |

No oversampled applied. Prevalence is computed at the clinical note level. For the same number of sepsis cases, the clinical note occurrences are different for a different time window. SERA algorithm uses the voting algorithm; dagging, and GBT algorithms are presented for comparative purposes.

least 0.80 sensitivity, specificity, and PPV values. The additional hours for early prediction is crucial in improving clinical outcomes as every 1 h delay in antimicrobial administration is associated with a decrease in patient's survival of 7.6%[6]. The additional 24 h lead time ahead of similar predictions will provide clinicians additional lead time to prepare for highly time-sensitive sepsis management intervention.

**SERA algorithm vs. human prediction**. We compared the performance of the SERA algorithm with other predictive scoring systems in the clinical context by juxtaposing the algorithm's prediction with the standard clinical practice of diagnosing sepsis or predicting mortality due to infection. Clinical practice typically utilizes standardized scoring systems such as systemic inflammatory response syndrome (SIRS), sequential organ failure assessment (SOFA), quickSOFA (qSOFA), and modified early warning system (MEWS) to predict sepsis or mortality due to infection. Based on the meta-analysis study[8], the typical true positive rate (TPR, sensitivity) of these four scoring systems have AUC ranging from 0.50 to 0.78, TPR ranging from 0.56 to 0.8, and false-positive rate (FPR, 1—specificity) ranging from 0.16 to 0.50 at 4 h before the onset of sepsis. We plotted the ROC curves of our early prediction algorithm against these reported TPR and FPR scores (Fig. 2a).

Further, we also estimated the TPR and FPR of the hospital physician's diagnosis of sepsis for all the patients' encounters within the independent, test sample. Based on medical guidelines for the management of sepsis[4], physicians are required to

normalize lactate in patients with elevated lactate levels for initial resuscitation. Further, physicians would, as part of the diagnosis, obtain appropriate routine microbiologic cultures before starting antimicrobial therapy in patients with suspected sepsis to minimize delay in the start of antimicrobials. Hence, according to the hospital practice, when a physician suspects sepsis, she is to request one or more microbiologic culture tests and lactate tests as per the above guidelines.

As such, in order to gauge the performance of the physician in predicting sepsis, we used the time when the physician orders both culture and lactate tests as the point in time when she suspects a patient has sepsis. The time between when orders are made and when the patient developed sepsis is determined as the prediction time window. Using this methodology, we computed the TPR and FPR for physicians for five time-windows—48, 24, 12, 6, and 4 h before the onset of sepsis. In Fig. 2a, we can see that at 4 h before the onset of sepsis, the SERA algorithm outperformed the hospital's physicians in predicting sepsis within the test sample. In addition, the SERA algorithm outperformed the typically reported accuracy rates of human-based scoring methods such as qSOFA, MEWS, SIRS, and SOFA as reported in prior studies[8].

Figure 2b presents the ROC of the early prediction algorithm for all the timings at 48, 24, 12, 6, and 4 h before sepsis. We observe that the ROC improves from 48 h to 12 h and maintains at a similar level beyond the 12-h mark. The ROC of the SERA algorithm at all time periods outperformed hospital physicians' predictions. As a conservative comparison, we contrasted SERA algorithm prediction at 48, 24, 12, 6, and 4 h against the

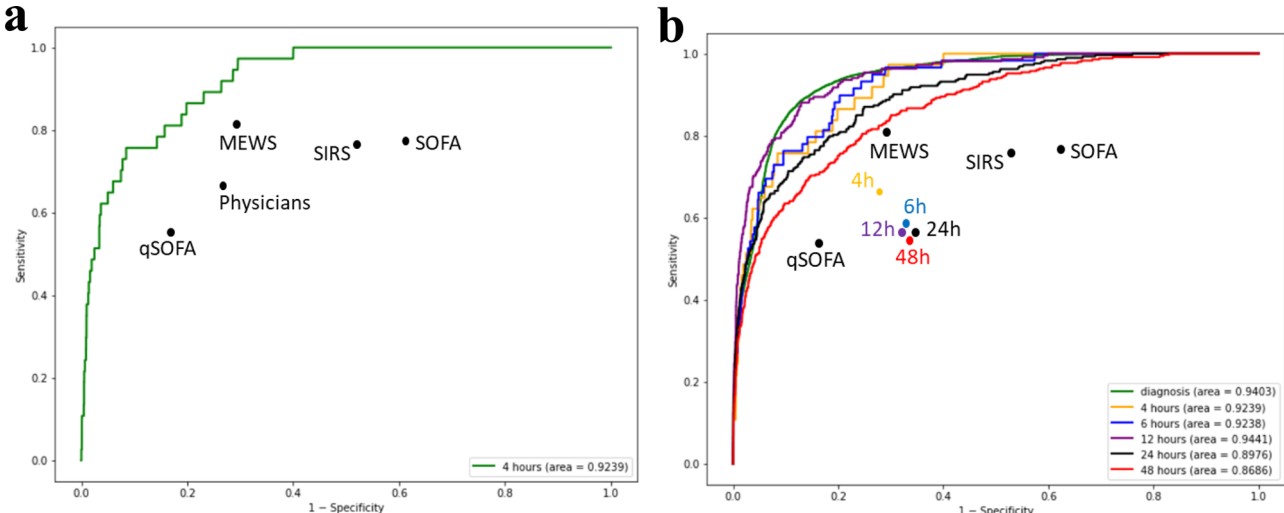

**Fig. 2 ROC curves for 48, 24, 12, 6, and 4-h early prediction. a**, **b** The ROCs represent the performance of early prediction algorithm at 4, 6, 12, 24, and 48 h prior to the onset of sepsis using the independent, test sample. "qSOFA", "MEWS", "SIRS", and "SOFA" represent the TPR and FPR from these methods employed by physicians in prior studies at 0–4 h prior to the onset of sepsis. "Physicians" represent TPR and FPR of patients in the independent, test sample set that were suspected by hospital's physicians to have sepsis at 4 h prior to the onset of sepsis. **b** "4 h", "6 h", "12 h", "24 h", and "48 h" represent TPR and FPR of patients in the independent, test sample set that were suspected by hospital's physicians to have sepsis at the respective time prior to the onset of sepsis.

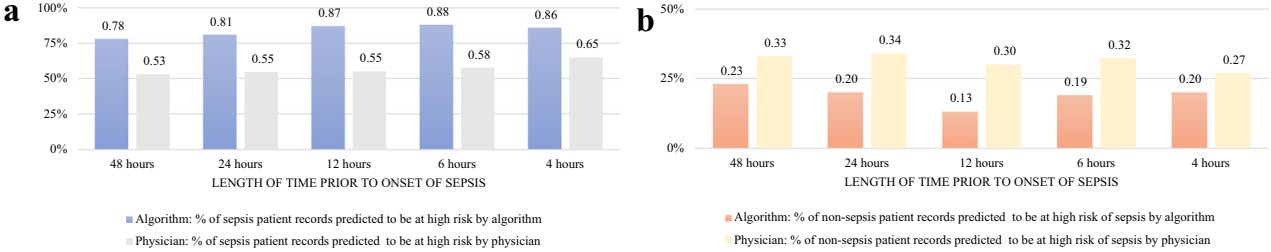

**Fig. 3 Comparing the performance of the SERA algorithm vs. physician. a** The bars represent the percentage of sepsis patient records correctly flagged as having a high risk of sepsis (likely to have sepsis) by either the algorithm or physicians. The chart compares the true positive rate of the algorithm's prediction at different lengths of time before the onset of sepsis against the true positive rate of physicians' prediction in the hospital. **b** The bars represent the percentage of non-sepsis patient records erroneously flagged as having a high risk of sepsis (likely to have sepsis) by either the algorithm or physicians. The chart compares the false-positive rate of the algorithm's prediction at different lengths of time before the onset of sepsis against the false-positive rate of physicians' prediction in the hospital.

prediction accuracy of MEWS, qSOFA, SIRS, and SOFA, performed at the 4-h mark, as reported in the literature[8]. We observe that SERA algorithm prediction for all four periods generally outperformed MEWS, qSOFA, SIRS, and SOFA prediction scores even when they are assessed at the 4-h mark before sepsis.

To examine the practical utility of the early prediction algorithm, we compute TPR and FPR of the early prediction algorithm at 48, 24, 12, 6, and 4-h mark (Fig. 3). We observed that 48-h from the onset of sepsis, the SERA algorithm can pick up 0.78 of all patients that eventually have sepsis, and this prediction (TPR) improves to at least 0.86 for time periods less than 12 h before the onset of sepsis. Hospital physicians, on the other hand, can detect approximately only 0.53 of all patients that eventually have sepsis at the 48-h mark, and this proportion only increases marginally to 0.58 at the 6-h mark. At the 4-h mark, hospital physicians do observe a measurable increase in TPR to 0.65. For all five periods, the SERA algorithm has a higher TPR of predicting sepsis by 0.21–0.32 compared to hospital physicians in the same period. This suggests that the SERA algorithm has the potential to increase the number of early sepsis detection by 21–32% compared to relying only on hospital physicians' assessment.

The FPR of the SERA algorithm varies from 0.23 at the 48-h mark to 0.13 at the 12-h mark (Fig. 3b). The FPR of hospital physicians' predictions, however, is considerably higher than the SERA algorithm, and it varies between 0.34 and 0.27 from 48-h to 4-h mark. Thus, the use of the SERA algorithm has the potential to reduce false positives by 0.07–0.17. Figure 3 suggests that the TPR and FPR of hospital physicians' peaked only at the 4-h mark—this means that physicians have a much shorter lead time for medical intervention. Conversely, the SERA algorithm is able to achieve considerably earlier prediction rates of up to 48 h prior to sepsis onset, which means that physicians have much earlier warnings of sepsis onset and thus more time for more effective interventions.

**Benefits of unstructured clinical text in early sepsis prediction**. To quantify the added predictive value of unstructured clinical notes in the SERA algorithm, we use only the structured variables, as seen in Table 1, to diagnose and predict sepsis. We then compare the algorithm's performance (with only structure variables) against the algorithm with structured and unstructured clinical notes (Fig. 4).

On the one hand, we observed that while the added use of clinical text did improve the performance of the SERA algorithm

## AUC

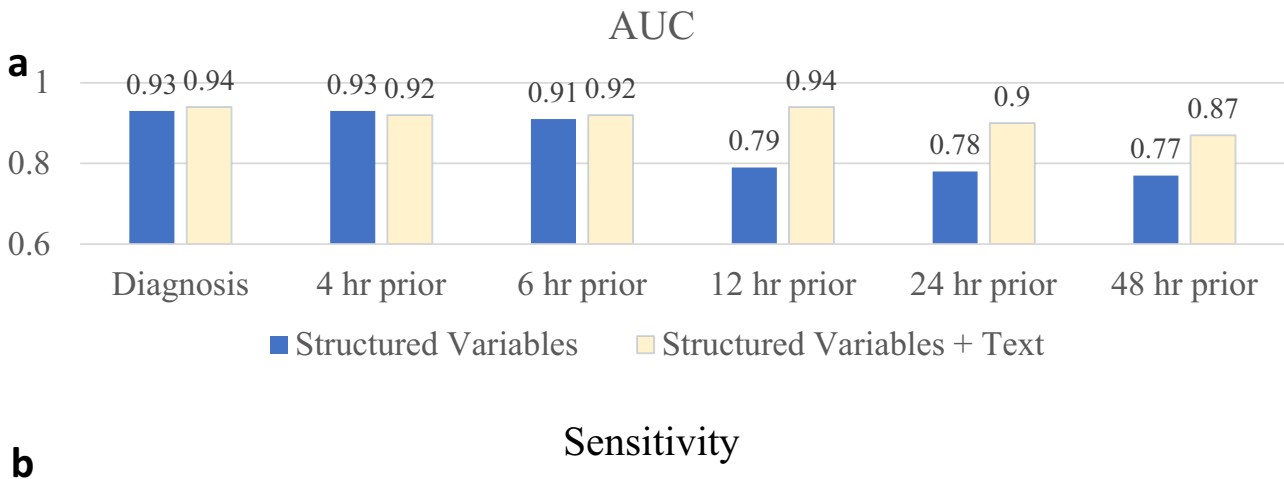

## Sensitivity

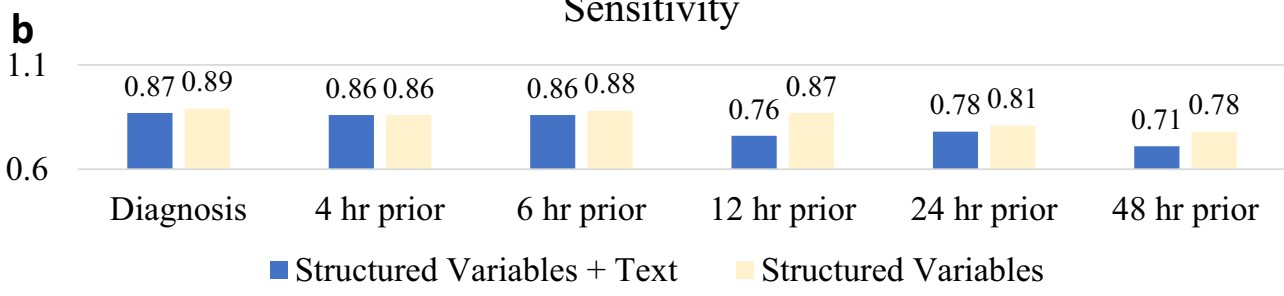

## Specificity

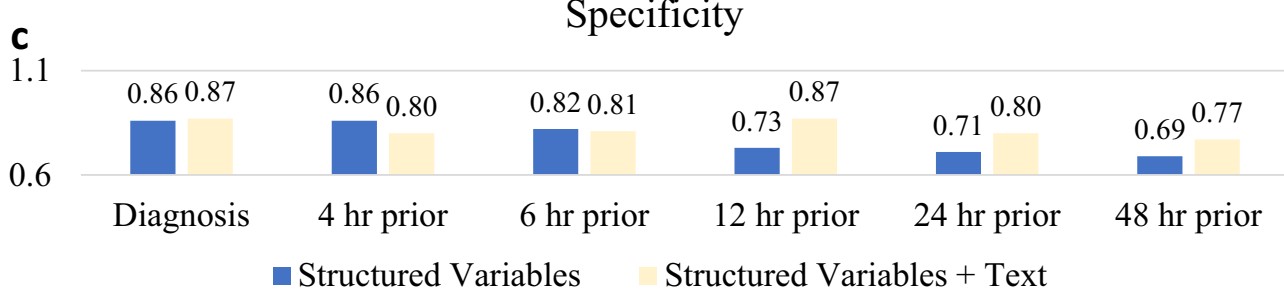

**Fig. 4 Comparing SERA algorithm performance with models without clinical text.** We measure the change in performance for each diagnosis and prediction model when clinical text is added as additional predictors. Structured variables represent the models that only include vitals, investigations, and treatment data as predictors. Structured variables + text represent the models have additional clinical notes as predictors. **a** Compares the AUC of models with and without clinical notes. **b** Compares the sensitivity of models with and without clinical notes. **c** Compares the specificity of models with and without clinical notes.

for diagnosis and early prediction (up to 6 h), the improvement was marginal. On the other hand, when we consider time frames between 12 and 48 h ahead of sepsis, we noted that the use of clinical text for early prediction had considerable benefits. Specifically, we observed that for the SERA algorithm in the 12–48-h early prediction window, the added clinical text improved (1) the AUC by between 0.10 and 0.15, (2) the sensitivity by 0.07–0.13, and (3) the specificity by 0.08–0.14.

The results suggest that for time periods closer to the onset of sepsis, the measurable symptoms of sepsis are manifested in the structured variables such as a drop in blood pressure values. In this instance, the added use of clinical text provides only marginal predictive gains to the SERA algorithm as the structured variables captured most of the sepsis symptoms. The unstructured clinical text, however, provides greater value in prediction when we are focusing on early prediction between 12 and 48 h prior to sepsis as the patient has yet to manifest observable symptoms that can be measured by the structured sepsis variables. Our model suggests that the physician's judgment and qualitative inputs of

the patient's prognosis at that time provide additional crucial data that can be used for predicting sepsis.

**Application of SERA algorithm in low sepsis prevalence environment.** In a retrospective cohort study of 409 United States hospitals from 2009 and 2014, sepsis prevalence ranges from 1.8% to 12% of all hospitalized patients. The mean prevalence is 6%, and this remains relatively consistent over time[19]. Despite the naturally low prevalence of sepsis in clinical settings, most studies develop sepsis prediction algorithms using oversampled datasets with a significantly higher prevalence of up to 50%[7,9,11,12,26–31]. In our study, we develop the models for both the oversampled environment, as seen in prior research, and for situations where the prevalence is low, as seen in a typical clinical environment.

As reported in Table 3, the models developed with the typical low clinical prevalence context of 6.15% can achieve high sensitivity, albeit with naturally low PPV. For purposes of clinical application, we simulate how changes in the prevalence of sepsis

**Table 4 Simulated PPV at different sepsis prevalence levels.**

| PPV (95% CI) | | | | | | | |
|---|---|---|---|---|---|---|---|
| Algorithm | Prevalence | 1.8% | 6% | 12% | 20% | 30% | 50% |
| | Diagnosis | 0.11 (0.11–0.11) | 0.30 (0.29–0.31) | 0.48 (0.47–0.49) | 0.63 (0.62–0.63) | 0.74 (0.74–0.75) | 0.87 (0.87–0.87) |
| | 4 h | 0.11 (0.10–0.12) | 0.30 (0.28–0.33) | 0.48 (0.45–0.51) | 0.63 (0.60–0.66) | 0.75 (0.72–0.77) | 0.87 (0.86–0.89) |
| | 6 h | 0.09 (0.08–0.09) | 0.25 (0.22–0.27) | 0.41 (0.39–0.44) | 0.56 (0.54–0.59) | 0.69 (0.67–0.71) | 0.84 (0.82–0.85) |
| | 12 h | 0.08 (0.08–0.09) | 0.24 (0.23–0.25) | 0.40 (0.39–0.42) | 0.55 (0.54–0.57) | 0.68 (0.67–0.69) | 0.83 (0.82–0.84) |
| | 24 h | 0.07 (0.06–0.07) | 0.20 (0.19–0.21) | 0.35 (0.33–0.36) | 0.49 (0.48–0.51) | 0.63 (0.61–0.64) | 0.80 (0.79–0.81) |
| | 48 h | 0.06 (0.05–0.06) | 0.17 (0.16–0.18) | 0.30 (0.29–0.32) | 0.44 (0.43–0.46) | 0.58 (0.56–0.59) | 0.77 (0.75–0.77) |

1.8% (12%) represents the low (high) end of the typical sepsis prevalence observed in hospitals based on prior research 19. Six percent represents the average sepsis prevalence of hospitals in the United States. Other prevalence percentages are for comparison purposes as they represent oversampled prevalence percentages typically observed in prior research.

impact the PPV (Table 4). For example, for a 12-h early prediction window, hospitals experiencing the lower end of 1.8% sepsis prevalence will have an estimated algorithm PPV of 8.3% (95% CI: 0.89–8.73%). If the prevalence of sepsis increases to the higher end of 12%, the estimated PPV increases to 40.25% (95% CI: 38.94–41.58%). The simulated results illustrate the applicability of the SERA algorithm in natural clinical settings where the prevalence of sepsis varies depending on the type of clinical specialty and/or location of the institutions.

## Discussion
The SERA algorithm can potentially provide the early detection of sepsis that clinicians need to effectively intervene and manage sepsis patients. Medical practice and prior studies[4] have shown that early detection of sepsis is challenging, and sepsis patients can deteriorate rapidly; thus, every minute counts in the diagnosis of sepsis. We have shown the effectiveness of the SERA algorithm in that it is able to flag 21–32% more patients at risk of sepsis compared to hospital physicians in a clinical setting between 4 and 48 h before the onset of sepsis. This additional lead time in sepsis alert provides greater opportunities for physicians to commence treatment, thereby lowering mortality.

As in every medical alert system, false positive in the alert system is a cost to the healthcare organization. Unacceptable levels of false positives can lead to wastage of medical resources as physicians need to perform additional follow-up diagnosis and treatment. Further, frequent false positives in a clinical setting will lead to a loss of confidence in these alerts among attending physicians, who will, in turn, be less likely to act on future alerts. A high number of false alerts may also reduce their confidence in other similar best practice alerts embedded within EMR systems. The SERA algorithm, however, has achieved high TPR while maintaining a reasonable low FPR—an FPR considerably lower than hospital physicians' early assessment of sepsis. Further, based on the simulation of different sepsis prevalence levels, we show that the PPV of our model is also appropriate for clinical applications, where the prevalence of sepsis is naturally low.

This study thus presents the potential of using AI as a "canary in a coal mine" for clinicians. We could envision running the SERA algorithm in the background to continuously monitor patients and act as an early warning system for patients who are at risk of having sepsis. At the same time, using the SERA algorithm in this manner could also potentially flag sepsis patients and thus prevent them from being overlooked; a situation that is common in stressful hospital settings where physicians have to continually make multiple medical decisions and judgments while managing high patient loads. We have provided details of how to practically set up this system in the methods section.

Through this study, we provide further empirical evidence for the value of data from EMR systems. Notably, we show that the data from EMR systems can be applied for more advanced

healthcare applications. Unlike prior studies that utilized only structured EMR data, we demonstrated the value of using the vast amounts of unstructured data embedded in progress notes. For a typical EMR system, the volume of unstructured data dwarfs the volume of structured data, and we believe that the value from unstructured data could be unlocked in a systematic fashion using NLP (see below). Using it for sepsis prediction as per our study is one case in point. The NLP-enabled analysis would be naturally suited for managing other complexes, equivocal, and critical clinical challenges as these challenges would benefit from expert insights embedded within EMR's unstructured data.

Our study also extends the use of NLP in healthcare research. Within the medical informatics literature, NLP has been widely used to extract concepts, entities, and events, as well as their relations and associated attributes from free-text[33,34], given that free-text is a common form of data in EMR systems[10]. Our study suggests that the use of NLP could be extended by integrating NLP processed text with traditional machine learning tools to assist in diagnosing and early detection of diseases. Our NLP results suggest that clinicians' progress notes, as well as other clinical reports (e.g., radiologist reports) stored in the EMR systems, may contain patterns of themes and topics that can be extracted and used productively for clinical analysis. Mining these notes are especially useful in clinical settings where the symptoms of ailments are not easily observed in structured data captured in the EMR. Our study also highlights the value of mining and analyzing progress notes, notwithstanding the use of medical abbreviations and the idiosyncratic manner in which clinicians record their diagnoses and treatment plans.

Like all studies, there are limitations to our study. While we have validated and tested the models with an independent, test sample from a later time period, future research should conduct external validation in different hospital settings to improve the SERA algorithm and make our findings more robust.

In conclusion, if predictive algorithms similar to the SERA algorithm can be designed to access valuable structured and unstructured data patient data continuously, hospital systems could potentially provide a 24 by 7, continual monitoring of patients' condition, thereby improving the ability for early sepsis detection and intervention. Though the SERA algorithm can achieve higher sensitivity and specificity rates compared to some physician's diagnoses and prior machine learning algorithms, we believe its primary role is to complement, not substitute, the clinical team's existing work. We further argue that other similar NLP-enabled algorithms could be developed to augment healthcare workers' knowledge and improve their decision making[35].

## Methods
**Data sample**. National Healthcare Group's domain specific review board granted the ethics approval on 26 July 2018 (approval reference 2018/00455). Informed consent is not required under the Human Biomedical Research Act 2015 (Singapore). Approval was granted to extract de-identified patients' data from the EMR

**Table 5 Topic categories.**

| Category | Count | Definition |
|---|---|---|
| Clinical status | 28 | Routine updates of clinical conditions as well as diagnosis (e.g., vitals) excluding lab and radio-diagnostic tests |
| Communication | 3 | Communication between staff |
| lab test | 24 | Orders and reports of lab or radio-diagnostic test results |
| Non-clinical status | 2 | Routine updates of non-clinical conditions |
| Social relationship | 2 | Information about family and social aspects of patient |
| Symptom | 10 | Clinical symptoms |
| Treatment | 31 | Treatment procedure or medication prescribed as well as the status of the treatment/ medication |

The 100 topics are classified into seven different categories. The distribution of topics among categories is similar if 25, 50, 75, or 150 topics are extracted instead. Detailed results are available upon request from the corresponding authors.

system to build the predictive algorithm. MySQL 8.0 was used to extract the data from the Epic™ electronic medical record system. The sample consists of 5317 patients admitted from 2 April 2015 to 31 Dec 2017 with 114,602 clinical note entries. We separated this sample into a training and validation sample and an independent, hold-out, test sample. We used a tenfold cross-validation methodology to train and validate the model on the training and validation sample. In order to test the stability of the text topics extracted from our text-mining algorithm, we deliberately designed the hold-out test sample to include patients admitted from a later period (10 May 2017 to 31 December 2017). The training and validation sample consists of 3722 patients (80,162 clinical note entries) while the independent, hold-out test sample consists of 1595 patients with 34,440 clinical note entries. We use each patient consultation instance (clinical note) as the unit of analysis (see Supplementary Fig. 2 in Supplementary Information for more details about the data sample).

**Text mining technique**. In addition to the structured data (Table 1) used in the predictive model, we utilized clinical notes data that are unstructured free-form text. Before the unstructured free-form text could be analyzed and used as part of a predictive model, we first utilize LDA[36] to codify the unstructured free-form text into a numerical representation. LDA, like other topic modeling algorithms, is an unsupervised technique that empirically creates topics based on patterns of (co-) occurrence of words found in the analyzed documents. Through LDA we generated topics from the clinical notes that are represented by a vector with individual topic loadings corresponding to the concentration of discussion for a particular topic.

We processed the unstructured free-form text using the following five steps. First, under HIPAA guidelines and standards for anonymization of data (Health Insurance Portability and Accountability Act of 1996, HIPPA), we removed all possible identifiers within the clinical notes. Identifiers that we parsed out included: names, geographic subdivisions, zip codes, ID numbers, all elements of date, telephone/ fax numbers, vehicle numbers, device serial numbers, and email addresses. Second, we tokenized the text contained in the documents by removing all punctuation marks; lemmatizing the words by replacing words with their root form; applying part-of-speech tagging, and removing stop words, such as articles and prepositions. We also removed an extended list of medical-related stop words or phrases that are common in these texts but have no practical meaning. Examples of such are "report," "progress," "provide," "lab unit" (a complete list of the words and phrases that are excluded from topic modeling is available upon request from the authors). After these two steps, we created a term-document matrix, where rows represented the occurrence (a measure of frequency) of a term in the documents, and columns represented the documents.

Third, we applied a text filter to the processed text to reduce the number of terms by eliminating rare terms and weighting the frequencies of terms with multiple occurrences. Fourth, we ran the LDA topic clustering algorithm on the terms to determine the various topics. Based on extant literature[37], the number of topics that can be developed from the text is highly subjective and is often a function of the number of observations or the expected variety of the topics in the dataset. Using these guidelines, and as part of our robustness checks, we attempted five different iterations with 25, 50, 75, 100, and 150 topics. Subsequent analysis of the results yielded qualitatively similar results for all five iterations in the final predictive model; thus, in this paper, we reported the 100-topic model for brevity. Drawing on the 100-topic model, we presented the 100 topics to three researchers to independently classify them into seven topic categories (Table 5). All text-mining is processed using SAS Enterprise Miner 14.2.

Note that the step of developing the topics is required only during the model development and validation phase. When using the text-mining algorithm to assess new clinical notes, the 100 topics developed are used as the benchmark to help compute the fit between the new clinical note to these 100 topics. The high fit measure represents a high similarity between the new clinical notes with the benchmark topics. From a practical implementation perspective, we found that the computation of the fit values is not computationally intensive. For example, a test-case patient with a lengthy clinical note of 1806 words (the median length of the clinical notes in our sample is 840 words) requires only 0.17 s to estimate these fit

values using an Intel i7 processor 2.7 GHz, with 16.0G RAM in SAS Enterprise Miner 14.2.

**Machine learning algorithm**. Ensemble methods are machine-learning algorithms that utilize multiple classifiers to determine the predicted outcome by taking a (weighted) vote of their predictions. These methods often perform better than any single classifier[38,39]. There are several different ensemble methods, such as voting, bagging, stacking, and boosting.

In our main estimation, we use a voting ensemble. Voting is an ensemble machine learning model that combines the predictions from multiple other models (base classifiers). Here, we use two base classifiers: a stochastic gradient descent (SGD) based logistics regression and a random forest algorithm. Our combination rule is an average of probabilities, i.e., we calculate the average probability of the two base models as our voted probability.

The first base classifier, SGD, is an optimizing algorithm that seeks to minimize the error in prediction by learning iteratively from prior fitted estimates. The method iteratively draws random samples from the training sample to estimate the parameters of the model that is used to classify a patient as having sepsis or not having sepsis. It learns from each sampling iteration to determine the accuracy of the classification and adjust the parameter estimation until further improvements in prediction results are minimal.

For each iteration, the predicted parameter $\beta$ is calculated, and the model is updated using the following logistic equation:

$$\beta^{\text{new}} = \beta^{\text{old}} + \text{lr}(y - \hat{y})\hat{y}(1 - \hat{y})x \tag{1}$$

where $\beta$ is the optimized parameter, $lr$ is a learning rate, $y - \hat{y}$ is the prediction error for the model in a particular iteration in the training data, $\hat{y}$ is the prediction made by the coefficients, and $x$ is the input value. In our case, the input variables were a combination of the structured variables (as indicated in Table 1) and the topic loadings of each clinical note on the 100 topics we extracted in the text mining procedure.

The second classifier used here for voting is a random forest classifier, with the case of sepsis being the target variable. The probabilities of both classifiers are averaged out to arrive at the final probability used in our voting ensemble model.

**Alternative estimators**. For comparative purposes, we also used two different alternative estimators, namely, dagging and GBT.

Dagging is an ensemble method that has been widely used in existing machine learning literature when the data is "noisy," i.e., data with a large amount of additional irrelevant information. In dagging, we partition the training data sample into a set of disjoint stratified samples. We then select the base classifier logistic regression with SGD as described earlier for this procedure, and train the data using this base classifier within each of the disjointed samples. Next, the ensemble method applied the results from the trained copies of the base classifier to the validation data sample. We compute the average across all sub-samples, and the predicted result is based on a vote across all sub-samples.

GBT uses an ensemble of multiple trees to create more powerful prediction models for classification and regression. The key idea is to build a series of trees, where each tree is trained so that it attempts to correct the mistakes of the previous tree in the series. GBT involves three elements: a loss function to be optimized, a weak learner to make predictions, and an additive model to add weak learners to minimize the loss function. Decision trees are used as the weak learner in GBT and a gradient descent procedure is used to minimize the loss when adding trees. A fixed number of trees are added, or training stops once loss reaches an acceptable level or no longer improves on the validation dataset. All ensemble machine learning was conducted using the KNIME Analytics Platform (version 4.1.6).

**Other model diagnostics**. To observe the trade-offs between PPV (precision) and sensitivity (recall), refer to the precision–recall curves in Supplementary Fig. 3 of Supplementary Information. Similarly, the SERA algorithm's calibration curves can be found in Supplementary Fig. 4 of the Supplementary Information.

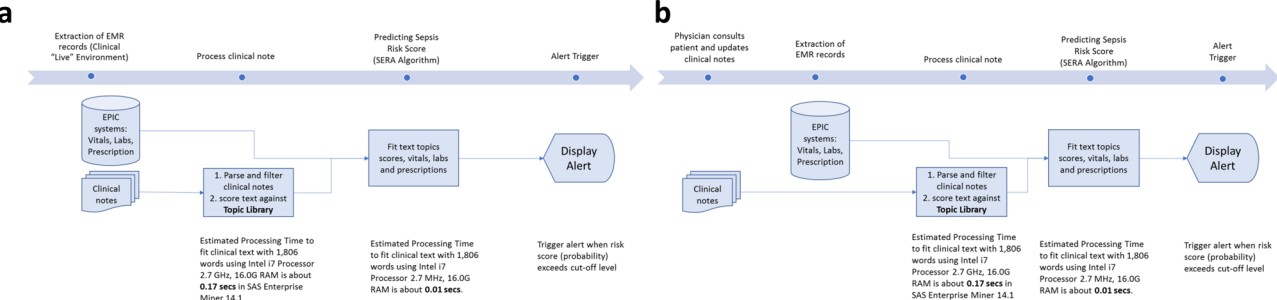

**Fig. 5 Workflow for the setup of SERA algorithm within the clinical environment. a** A proposed workflow for the SERA algorithm to operate in the background and provide alerts when key events (such as ward shift handovers) are triggered. **b** A proposed workflow for the SERA algorithm to run immediately after a physician updates a patient's clinical notes in the EMR system.

**Proposed setup of SERA algorithm within the clinical environment**. We propose two possible modes in which the SERA algorithm can operate within the clinical environment.

**Background mode**. In this first mode, the SERA algorithm is designed to run in the background. Specifically, it is configured to run during key events using the latest patient's clinical data available, e.g., during ward shift handovers. If the risk score exceeds the designated cut-off level, the EMR system will alert the physician. Alternatively, if there are more computing resources available, hospitals can choose to run it in fixed hourly-time intervals. For a large 500-bed hospital, if we run all cases together, the SERA algorithm will take ~90 s to score all 500 patients. This approach ensures an ongoing, regular time-based sepsis risk assessment for patients within the hospital. (See Fig. 5a for the workflow for this mode)

**Ad-hoc mode**. In the second mode, the algorithm can be designed to run immediately after a physician updates her clinical notes in the EMR system. In this case, the SERA algorithm will run in an ad-hoc manner since the score is only calculated after a physician has updated the patient's status. The SERA's score then acts as a decision support mechanism to flag suspected sepsis cases. As observed from the study, the SERA algorithm outperforms physicians in the early prediction of sepsis and thus may be an important early warning indicator to assist physicians in their diagnosis and patient care. (See Fig. 5b for the workflow)

**Reporting summary**. Further information on research design is available in the Nature Research Reporting Summary linked to this article.

## Data availability
The raw datasets generated during and analyzed during the current study will not be published publicly due to privacy regulations under the Human Biomedical Research Act (HBRA) 2015 (Singapore). Raw datasets are available for review purposes. The raw data consists of clinical data from patients, including textual clinical notes were written by physicians and contain information that could compromise research participant (patient) privacy or consent. The processed textual data with vitals is however available from the corresponding author on reasonable request.

## Code availability
The code used in the current study is largely based on the open-sourced software KNIME with some custom modifications, which will be made available from the corresponding authors upon reasonable request.

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

## Acknowledgements

This work is supported by the Ministry of Education, Singapore, under the Social Science Research Council Thematic Grant. Grant number: MOE2017-SSRTG-030.

## Author contributions

L.W. and K.H.G. were involved in the formal analysis and methodology of the paper. K.H.G., L.W., and A.Y. were involved in the writing of the paper. H.P., L.K., J.Y., and G.T. were involved in data identification, extraction, and curation. All authors were involved in the conceptualization of the project.

## Competing interests

The authors declare no competing interests.
