## [Peer Review File · Nature Communications]

Reviewers' Comments:

Reviewer #1:

Remarks to the Author:

As authors are aware, there has been quite a lot of work dedicated to both sepsis detection and prediction. As such, there should be a more extensive review of the work already carried out. The major claim of this work is that methods for sepsis diagnosis and prediction consider only structured data and not clinical notes. However, there are other studies that also have addressed this issue, most recently study in [1].

Based on the title of the paper "role of unstructured data", I would have expected the authors to describe the actual role of clinical notes in diagnosis/prediction performance, in comparison to using structured data only. However, the results presented pertain only to processing of both types of data together. Thus, it is not known if clinical notes play a role in increasing predictive performance.

I found the method of labelling sepsis prediction quite unusual and a potential source of significant confusion: "For each patient encounter, when a physician suspects sepsis, she will at least request a culture test and lactate test. Thus, when the physician orders for both tests, we classify the patient as one predicted to have sepsis by the physician,". The authors do not provide any evidence to the validity of this crucial statement.

The dataset is highly imbalanced; thus, AUC should not be the only performance metric reported. Authors should at least provide PPV and NPV. Furthermore, it is unclear how well calibrated the model is; thus, calibration curves should also be provided.

When developing the model, cross-validation is much more robust method to avoid overfitting than the random split used by the authors.

Given the above considerations and the amount of manual work required in annotation of LDA output, I find it difficult to see how this algorithm may enter clinical practice as outlined in the discussion section.

[1] R. Liu, J. L. Greenstein, S. V. Sarma and R. L. Winslow, "Natural Language Processing of Clinical Notes for Improved Early Prediction of Septic Shock in the ICU," 2019 41st Annual International Conference of the IEEE Engineering in Medicine and Biology Society (EMBC), Berlin, Germany, 2019, pp. 6103-6108. doi: 10.1109/EMBC.2019.8857819

Reviewer #2:

Remarks to the Author:

Thank you for the opportunity to review this interesting paper. I only have some minor suggestions that I think would help clarify the manuscript for the reader.

The background is well motivated. This reviewer whole-heartedly agrees in the use of ML for real-time surveillance, specifically in the area of workflow augmentation for applications such as decreasing variability in care, as the authors have eloquently stated in their introduction.

The authors might consider citing this article in the background <https://www.ncbi.nlm.nih.gov/pmc/articles/PMC5383046/> which is directly applicable to this work, which also shows that using unstructured data, in addition to structured data, substantially improves this prediction task. More importantly, this manuscript reports performance metrics

substantially better than this comparison paper (0.86 vs. 0.92), which is considerably, and would make a good discussion point, and in fact these performance characteristics persist even 12 hours prior.

In the Methods section, under Data Sample, how was random sampling performed? Was the unit of randomization performed at the level of the note, the patient visit, or the patient. Also, was there any overlap in patients between the test set and the training/validation set?

In the methods section, it would be helpful if the ICD-10 codes for cohort selection were explicitly mentioned.

Methods: processing of clinical notes: Please cite the implementation (software package) that was used, as well as for your prediction model.

The paper is missing a demographics table to describe the patient population. For example, what is the incidence of sepsis, severe sepsis, septic shock? How many are admitted to the ICU? What is the age distribution?

What was the class imbalance of your prediction and how did you account for it? It is unclear from the manuscript if a balanced dataset was created by randomly undersampling the non-sepsis cohort, or if the class imbalance was dealt with in some other manner during training.

It would be helpful if a standard CONSORT enrollment diagram was included as a figure, potentially to replace Figure 1.

It would also be helpful if one included a reliability diagram (calibration diagram) as well as a precision-recall diagram for a representative model to better understand calibration as well as the trade-offs between precision and recall for choosing a decision threshold.

Response to Reviewer 1

Reviewer #1 (Remarks to the Author):

As authors are aware, there has been quite a lot of work dedicated to both sepsis detection and prediction. As such, there should be a more extensive review of the work already carried out.

The major claim of this work is that methods for sepsis diagnosis and prediction consider only structured data and not clinical notes. However, there are other studies that also have addressed this issue, most recently study in [1].

Thank you for pointing out these studies that examined sepsis diagnosis and prediction. In this revision, we have incorporated them in our literature review. Specifically, we have cited and acknowledged the strengths of Liu, Greenstein, Sarma, and Winslow (2019) 's sepsis detection algorithm. As per Senior Editor's request to discuss how our work compares with Liu et al. 's (2019) on sepsis detection and prediction algorithm, we would like to highlight the following key points:

1. Strength and robustness of our prediction algorithm

- **Our algorithm provides earlier prediction of sepsis up to 24 hours.** Whereas Liu et al. (2019) 's algorithm for early prediction of sepsis is given at 7 hours prior to the onset of sepsis (AUC: 0.92, Sensitivity: 0.84, Specificity: 0.82), our algorithm's early sepsis prediction is effective up to 24 hours before the onset. Furthermore, our algorithm's results at 12 hours before the onset is comparable to Liu et al. (2019) 's algorithm at 7 hours (i.e., our 12 hour AUC: 0.94, sensitivity:0.87 and specificity:0.87). Given that studies have found that one hour delay in antimicrobial administration for sepsis patients is associated with a decrease in survival of 7.6% (Kumar et al., 2006), the ability to provide an sepsis alert five hours ahead of onset would substantially increase a patient's survival.
- **Our algorithm works in natural clinical setting's level of prevalence.** The dataset we used to *develop* the model was extracted from a natural clinical environment. We *tested* the algorithm in a dataset where the *prevalence* of sepsis is low (only **6.15% of all the patients** in the sample have sepsis). This level of prevalence is equivalent to the level typically observed in hospitals. We confirmed this with Rhee et al. (2017) study that examined 7.8 million patients in 409 different US hospitals from 2009 to 2014. They found that the prevalence of sepsis is about 6% of the patient population in those hospitals and that the level of prevalence was relatively stable across time. (cf. pp. 1246 (Rhee et al., 2017)). As such, this shows the efficacy of our algorithm in a natural clinical environment (more details of the workflow later). When compared to Liu et al. 2019's study that had a level of prevalence of approximately 41.2% (15,930/38,645), our algorithm was still able to achieve a high AUC despite the (low) level of prevalence.
- **Oversampling for comparsion.** Finally, in order to compare directly with Liu et al. 's (2019) study, we analyzed our model under high prevalence levels as per Liu et al. 's (2019) study. Specifically, we applied SMOTE (Synthetic Minority Over-sampling Technique) to our model (as suggested by R2). SMOTE oversampling has been applied in other studies – published in *Nature Communications* – developing machine learning classifiers for oral cancer detection (Carnielli et al., 2018) and cell identification/ classification (Rennie, Dalby, van Duin, & Andersson, 2018; Xia et al., 2020). These studies are similar to our study in that they also face low prevalence of positive cases. As seen from Table 1, we were able to achieve similar diagnostic stats

under such conditions.

Table 1: Using Voting Ensemble Algorithm (SMOTE)

Hours Prior to Sepsis	AUC	Sensitivity	Specificity	PPV
48	0.8686	0.7819	0.7673	0.7700
24	0.8976	0.8089	0.7972	0.8000
12	0.9441	0.8716	0.8741	0.8737
6	0.9238	0.8814	0.8070	0.8202
4	0.9239	0.8649	0.8015	0.8133

2. We compared our prediction algorithm with human physicians

- **Comparing algorithm’s performance with human physician.** In our study, we compared the performance of our algorithm against human physician’s early detection of sepsis cases. We found that our algorithm out-performs human in early sepsis detection up to 48 hours ahead of sepsis onset. Although the algorithm by Liu et al. (2019) is able to achieve high AUC at 7 hours ahead of sepsis onset, they did not report any attempts to compare their algorithm with human physicians.

3. Our prediction algorithm uses a more stable NLP technique

- **Stability of Natural Language Processing (NLP) – words vs topics.** Liu et al. (2019) employed a text modeling technique that extracts commonly used words in the clinical notes as predictors of sepsis. In our paper, we extend this technique by aggregating topics from those words extracted from clinical notes. Each topic is characterized by a collection of words that cluster around a common theme. We then employed these topics as predictors for sepsis. This NLP technique is better in the following ways:
 - **Stability over time.** Lexicographical topics are more stable compared to individual words (Blei, 2012; Wallach, 2006). Synonyms or phrases which carry similar meanings can be substituted to characterise a particular topic or phenomenon which falls within the same topic. Individual words on the other hand have narrower meanings and are unable to capture the use of synonyms. To show that our topics are stable over time, we tested our model on a test sample that included patients who were admitted at a later time.
 - **More accurate and generalizable.** When we compare classifiers that are built using topic features to those that are built using word features, the former classifiers were found to be more accurate (Blei, 2012; Blei, Ng, & Jordan, 2003; Wallach, 2006). This is because individual writers (e.g., clinicians) have idiosyncratic writing styles that may influence their choice of words. By using topics to extract and process notes, we can mitigate some of idiosyncratic writing styles and words employed by different writers. As a result, we believe this technique provides a NLP structure (topics) that is more generalizable across context and domains (e.g., different hospitals or specialty). We will describe below how we can operationalize this process in a clinical context (see our response below).

Based on the title of the paper “role of unstructured data”, I would have expected the authors to describe the actual role of clinical notes in diagnosis/prediction performance, in comparison to using structured data only. However, the results presented pertain only to

processing of both types of data together. Thus, it is not known if clinical notes play a role in increasing predictive performance.

Thank you for pointing out this fact. In this revision, we have provided the comparison between the model that used only structured variables (e.g. vitals) and the model that used both structured variables and clinical notes. From Table 2, we can see that the model that used both structured variables and clinical notes is significantly more accurate than the structured variable for models more than 12 hours prior to the onset of sepsis. For time periods less than 12 hours, the structured variables provide relatively good prediction as seen in prior literature, but in this research we show that unstructured clinical notes play a more important role in predicting sepsis more than 12 hours prior to the onset of sepsis.

Table 2A: Comparing Models with NLP (Without SMOTE)

Model	Structured Variables		Structured + NLP	
	AUC	Sensitivity Specificity	AUC	Sensitivity Specificity
Diagnosis	0.9282	0.8777 0.8612	0.9421	0.8854 0.8686
Prediction (T= 4 hours prior)	0.9051	0.8649 0.8236	0.9469	0.8919 0.8698
Prediction (T= 6 hours prior)	0.9134	0.8814 0.8309	0.9332	0.8814 0.8301
Prediction (T= 12 hours prior)	0.8102	0.7477 0.7203	0.9402	0.8807 0.8217
Prediction (T= 24 hours prior)	0.7765	0.7372 0.6882	0.8987	0.8089 0.7929
Prediction (T= 48 hours prior)	0.7479	0.6601 0.6925	0.8616	0.7649 0.7604

Table 2B: Comparing Models with NLP (With SMOTE)

Model	Structured Variables		Structured + NLP	
	AUC	Sensitivity Specificity	AUC	Sensitivity Specificity
Diagnosis	0.9278	0.8679 0.8627	0.9403	0.8853 0.8694
Prediction (T= 4 hours prior)	0.9343	0.8649 0.8613	0.9239	0.8649 0.8015
Prediction (T= 6 hours prior)	0.9080	0.8644 0.8167	0.9238	0.8814 0.8070
Prediction (T= 12 hours prior)	0.7889	0.7569 0.7274	0.9441	0.8716 0.8741
Prediction (T= 24 hours prior)	0.7755	0.7782 0.7149	0.8976	0.8089 0.7972
Prediction (T= 48 hours prior)	0.7695	0.7139 0.6874	0.8686	0.7819 0.7673

I found the method of labelling sepsis prediction quite unusual and a potential source of significant confusion: “For each patient encounter, when a physician suspects sepsis, she will at least request a culture test and lactate test. Thus, when the physician orders for both tests, we classify the patient as one predicted to have sepsis by the physician.” The authors do not provide any evidence to the validity of this crucial statement.

Thank you for this clarification question. We agree that our statement was unclear and thus may have created some confusion for the reader. We would like to clarify that:

- First, our classification for sepsis vs. non-sepsis cases is based on the ICD-10 classification. The ICD-10 classification is used for the training/ validation of the model as well as for the verification of the test results. The list of ICD-10 codes is now presented in the paper as requested by R2.

- Second, the statement quoted above refers to the way we measured the event where a ***hospital physician suspects*** a patient has sepsis (i.e., physician’s prediction of sepsis). We measured such events to compare the performance of our ***early prediction algorithm*** with physicians’ performance in predicting sepsis. This measure was NOT used to train/validate or test the model in any way. The measure was simply to ***determine human physicians’ performance*** of predicting sepsis.
- Third, the two criteria described here to determine the physician’s prediction of sepsis (i.e., request for lactate test and a culture test) are based on international guidelines for sepsis management and is part of the hospital’s operating procedures (see below).
 - Based on the International Guidelines for Management of Sepsis and Septic Shock: 2016 (Rhodes et al., 2017), as part of the guidelines for ***initial resuscitation***, physicians are required to ***normalize lactate in patients*** with elevated lactate levels as a marker of tissue hypoperfusion. As part of ***diagnosis***, the international guidelines also “recommend that ***appropriate routine microbiologic cultures*** (including blood) be obtained before starting antimicrobial therapy in patients with suspected sepsis or septic shock if doing so results in no substantial delay in the start of antimicrobials (BPS).” pp. 312 (Rhodes et al., 2017). As such culture and lactate tests are among the first two tests to be conducted when a patient is suspected to have sepsis.
 - We verified with the hospital management that the standard operating procedure when a physician suspects a patient has sepsis is to request for at least one culture test together with a lactate test.

The dataset is highly imbalanced; thus, ROC should not be the only performance metric reported. Authors should at least provide PPV and NPV. Furthermore, it is unclear how well calibrated the model is; thus, calibration curves should also be provided.

Thank you for the suggestion. In this revision, we have provided PPV, NPV, and the calibration curves. As noted earlier, our dataset is imbalanced and as suggested by the Senior Editor and R2 in this revision, we used SMOTE to oversample the positive cases to develop a more balanced dataset while training the model. A few points to highlight in this revision.

The unit of analysis for our prediction model is each single entry of clinical note – this ***is the same*** as per our initial submission. We define this as the unit of analysis because every instance the physician assesses the patient and inputs the clinical notes, she is making a clinical judgment. Hence, this unit of analysis is the most realistic in clinical setting and any sepsis alert should be presented at this point in time.

In clinical settings, sepsis has naturally low occurrence of 2% incidence rate per year with about 6% prevalence (Rhee et al., 2017). As PPV is directly related to the prevalence of sepsis (see equation 1 below), we expected low PPV values given the low prevalence in our data sample. To create a balanced sample (i.e. higher prevalence of sepsis), some machine learning studies under-sample the non-sepsis cases (Liu et al., 2019). But to build predictive models for classification tasks in a medical context, some researchers have argued that oversampling (instead of undersampling) can result in more accurate models (Batista, Prati, & Monard, 2004; Carnielli et al., 2018; Chawla, Bowyer, Hall, & Kegelmeyer, 2002). This method is used in studies that develop machine learning classifiers in low prevalence environment, e.g., oral cancer detection (Carnielli et al., 2018) and cell identification/classification (Rennie et al., 2018; Xia et al., 2020). As such, given these studies as well as the fact that under-sampling is not a viable option given the naturally, low prevalence of sepsis, we chose to oversample the sepsis cases using SMOTE (Synthetic Minority Over-sampling Technique).

The tables below show the diagnostics of the models (AUC, sensitivity, specificity, PPV, NPV with corresponding prevalence value). To check against overfitting – which is a criticism of oversampling – we also report our models without SMOTE for comparison. We are glad to report that other than PPV, the AUC, sensitivity, and specificity are equivalent for both oversampled and non-oversampled models.

Models that were run in natural environments of low prevalence ***without any oversampling*** are labelled as “Original data” and models with higher prevalence achieved by oversampling are labelled as “Smote #%” where # represents the extent of SMOTE. For example, SMOTE to 10% represents oversampling the sepsis cases up to the point where the sepsis cases make up to 10% of the overall sample. As prior literature suggest a “SMOTE to 50%” approach, as a robustness check, we provided five different levels of SMOTE for the early prediction models presented below.

The ***unit of analysis*** for the algorithm is ***each clinical note*** entry by the physician. As such, the prevalence figures presented in Table 3a to 3f below are computed ***at the clinical note*** level and not at the patient-encounter level. The original data prevalence figures (at the clinical note level) vary due to differences in time windows. Although the number of sepsis cases remains the same for the test sample across different time windows, the number of sepsis notes reduces with shorter time windows. Since the number of non-sepsis cases (and notes) remains the same in the test sample, this eventually leads to a reduction in prevalence with shorter time windows.

$$PPV = \frac{\text{sensitivity} \times \text{prevalence}}{(\text{sensitivity} \times \text{prevalence}) + [(1 - \text{specificity}) \times (1 - \text{prevalence})]} \quad \text{Eq. 1}$$

Table 3a: Diagnostic Model

		Vote Algorithm					GBT	Dagging
Test Data	Prevalence	AUC	Sensitivity	Specificity	PPV	NPV	AUC	AUC
Original data	17.69%	0.9421	0.8854	0.8686	0.5916	0.9724	0.9399	0.9234
Smote to 50%	46.23%	0.9403	0.8853	0.8694	0.8535	0.8981	0.9359	0.9188

Table 3b: Up to 4 hours (before Sepsis)

		Vote Algorithm					GBT	Dagging
Test Data	Prevalence	AUC	Sensitivity	Specificity	PPV	NPV	AUC	AUC
Original data	0.131%	0.9469	0.8919	0.8698	0.0089	0.9998	0.9387	0.9198
Smote to 10%	9.998%	0.9439	0.8649	0.8351	0.3682	0.9823	0.9290	0.8766
Smote to 20%	19.976%	0.9372	0.8649	0.8309	0.5608	0.9610	0.9308	0.8656
Smote to 30%	30.005%	0.9287	0.8649	0.8286	0.6839	0.9347	0.9170	0.8533
Smote to 50%	49.996%	0.9239	0.8649	0.8015	0.8133	0.8557	0.9165	0.8476

Table 3c: Up to 6 hours (before Sepsis)

Test Data	Prevalence	Vote Algorithm					GBT	Dagging
		AUC	Sensitivity	Specificity	PPV	NPV	AUC	AUC
Original data	0.208%	0.9332	0.8814	0.8301	0.0107	0.9997	0.9277	0.9015
Smote to 10%	9.954%	0.9401	0.8983	0.8531	0.4034	0.9870	0.9391	0.9037
Smote to 20%	20.018%	0.9360	0.8814	0.8378	0.5763	0.9658	0.9356	0.9015
Smote to 30%	29.951%	0.9283	0.8814	0.8180	0.6743	0.9416	0.9349	0.8960
Smote to 50%	49.976%	0.9238	0.8814	0.8070	0.8202	0.8719	0.9230	0.8912

Table 3d: Up to 12 hours (before Sepsis)

Test Data	Prevalence	Vote Algorithm					GBT	Dagging
		AUC	Sensitivity	Specificity	PPV	NPV	AUC	AUC
Original data	0.77%	0.9402	0.8807	0.8217	0.0369	0.9989	0.9272	0.9159
Smote to 10%	9.788%	0.9475	0.8761	0.8522	0.3914	0.9845	0.9332	0.9302
Smote to 20%	19.872%	0.9476	0.8670	0.8662	0.6164	0.9633	0.9316	0.9278
Smote to 30%	29.886%	0.9455	0.8670	0.8665	0.7346	0.9386	0.9258	0.9246
Smote to 50%	49.994%	0.9441	0.8716	0.8741	0.8737	0.8719	0.9193	0.9220

Table 3e: Up to 24 hours (before Sepsis)

Test Data	Prevalence	Vote Algorithm					GBT	Dagging
		AUC	Sensitivity	Specificity	PPV	NPV	AUC	AUC
Original data	1.0%	0.8987	0.8089	0.7929	0.0392	0.9975	0.8896	0.8814
Smote to 10%	10.3%	0.9002	0.8157	0.7919	0.3105	0.9740	0.8924	0.8739
Smote to 20%	20.0%	0.9002	0.8055	0.8049	0.5086	0.9429	0.8854	0.8666
Smote to 30%	30.0%	0.9012	0.8089	0.8048	0.6395	0.9077	0.8805	0.8672
Smote to 50%	50.1%	0.8976	0.8089	0.7972	0.8000	0.8062	0.8617	0.8642

Table 3f: Up to 48 hours (before Sepsis)

Test Data	Prevalence	Vote Algorithm					GBT	Dagging
		AUC	Sensitivity	Specificity	PPV	NPV	AUC	AUC
Original data	1.2%	0.8616	0.7649	0.7604	0.0387	0.9961	0.8511	0.8232

Smote to 10%	10.2%	0.8693	0.7960	0.7720	0.2838	0.9709	0.8628	0.8280
Smote to 20%	20.1%	0.8745	0.7847	0.7833	0.4773	0.9352	0.8537	0.8284
Smote to 30%	30.0%	0.8661	0.7762	0.7676	0.5888	0.8889	0.8424	0.8265
Smote to 50%	49.9%	0.8686	0.7819	0.7673	0.7700	0.7793	0.8292	0.8263

Figure R1: Calibration Curves for Diagnosis and Early Prediction Models

When developing the model, cross-validation is much more robust method to avoid overfitting than the random split used by the authors.

Thank you for your suggestion. In this revision, we used 10-fold cross validation modelling to prevent overfitting. The results obtained were similar to our initial submission.

Given the above considerations and the amount of manual work required in annotation of LDA output, I find it difficult to see how this algorithm may enter clinical practice as outlined in the discussion section.

Thank you for highlighting this point. We agree with you that the practical application of this algorithm is critical to the usefulness of the algorithm.

1. Before we discuss the practical use of the algorithm, we would like to first clarify the aspect of building and using the LDA output. Although training the topic library might be time intensive, this is only performed during the initial development of the topic library. Once the topic library is developed, it will be deployed to score new clinical text. As such, the topic library construction is only performed once at the

beginning of the project. As shown in the testing of our model, the topic library can effectively predict sepsis cases using clinical notes that were entered six months later. Subsequently, we only need to periodically update the topic library, which can be done using an automated workflow routine (e.g., using SAS Enterprise Miner).

2. We propose the following steps to run the SERA algorithm in a clinical setting for a patient:
 - a. Clinical note scoring process – scoring of a **new** clinical text in clinical setting involves three steps:
 - i. Parsing: tokenization, lemmatisation, and POS tagging
 - ii. Filtering: to weight terms
 - iii. Topic assignment (based on existing topic library that had been earlier developed)

The duration of computation to process and score the text is relatively short. To illustrate, we use a test-case patient with a long clinical note of 1,806 words (the median length of clinical note in our sample is 840 words). The clinical note scoring process using an Intel i7 Processor 2.7 GHz, 16.0G RAM is about **0.17 secs** in SAS Enterprise Miner 14.1

- b. SERA algorithm score process – after the clinical text is processed and scored, we will combine that with the structured variables from the EMR system and predict the likelihood of sepsis using the SERA algorithm. Here, the estimate processing time for all inputs using Intel i7 Processor 2.7 MHz, 16.0G RAM is about **0.01 secs**.

Together, the total duration to fit a new patient’s data to the SERA algorithm takes about **0.18 secs** from the moment the data is made available in the system.

3. The SERA algorithm can work in two different modes within the clinical environment.
 - a. Background mode: In this first mode, the algorithm is designed to run in the background. Specifically, it is configured to run at key events using the latest patient’s clinical data available, e.g., during ward shift handovers. If the risk score exceeds the designated cut-off level, the physician will be alerted via the EMR. Alternatively, if there are more computing resources available, hospitals can choose to run it in fixed hourly-time intervals. For a large 500-bed hospital, assuming if the algorithm runs the cases individually, it will approximately take 90 secs to completely score all 500 patients. This approach ensures an ongoing, regular time-based sepsis risk assessment for patients within the hospital. (See Figure R2 on the workflow for this mode)
 - b. Ad hoc mode: Second, the algorithm can be designed to immediately run after a physician submits her clinical notes in the EMR system. In this case, the SERA algorithm is run in an ad-hoc manner since the score is only applied after a physician has updated the patient’s status. The algorithm’s score then acts as a decision support to flag out suspected sepsis cases. As observed from study, the SERA algorithm outperforms physicians in early prediction of sepsis and thus may be an important early warning indicator for physicians to take note. (See Figure R3 for the workflow)

Figure R2: Workflow for Continual Time-based Sepsis Monitoring System (cSERA)

Figure R3: Workflow for Ad-Hoc Sepsis Monitoring System (aSERA)

[1] R. Liu, J. L. Greenstein, S. V. Sarma and R. L. Winslow, "Natural Language Processing of Clinical Notes for Improved Early Prediction of Septic Shock in the ICU," 2019 41st Annual International Conference of the IEEE Engineering in Medicine and Biology Society (EMBC), Berlin, Germany, 2019, pp. 6103-6108. doi: 10.1109/EMBC.2019.8857819

Response to Reviewer 2

Reviewer #2 (Remarks to the Author):

Thank you for the opportunity to review this interesting paper. I only have some minor suggestions that I think would help clarify the manuscript for the reader.

The background is well motivated. This reviewer whole-heartedly agrees in the use of ML for real-time surveillance, specifically in the area of workflow augmentation for applications such as decreasing variability in care, as the authors have eloquently stated in their introduction.

The authors might consider citing this article in the background <https://www.ncbi.nlm.nih.gov/pmc/articles/PMC5383046/> which is directly applicable to this work, which also shows that using unstructured data, in addition to structured data, substantially improves this prediction task. More importantly, this manuscript reports performance metrics substantially better than this comparison paper (0.86 vs. 0.92), which is considerably, and would make a good discussion point, and in fact these performance characteristics persist even 12 hours prior.

Thank you for taking the time to review our paper and pointing out the reference (Horng et al., 2017). We have cited it in this revision and incorporated the points, where relevant, in our revision. We hope that in this revision, we have resolved the issues you raised.

In the Methods section, under Data Sample, how was random sampling performed? Was the unit of randomization performed at the level of the note, the patient visit, or the patient? Also, was there any overlap in patients between the test set and the training/validation set?

The unit of sampling was performed at the patient visit level. The sample period is from 1st April 2015 to 31 Dec 2017 (first patient record 2nd April 2015). All sepsis patients (based on ICD-10 classification) were included in the dataset and we randomly selected non-sepsis patients to make up the rest of the sample. Given that the training/validation dataset is from an earlier set of patients and the test set is on a later set of patients, we had an overlap of 1 sepsis patient (out of 327 encounters) and 241 non-sepsis patients (out of 4990 encounters). It is important to note that while the patients are same, the notes were for **different** hospitalization encounters.

We have included a modified STROBE/CONSORT diagram as requested in this review for easier representation as well.

In the methods section, it would be helpful if the ICD-10 codes for cohort selection were explicitly mentioned.

ICD-10 codes used were:

- **SEPSIS:** 'A40.0','A40.1','A40.8','A40.9','A41.2','A41.0','A41.0Z16','A41.1','A40.3', 'A41.4','A41.50','A41.3','A41.51','A41.52','A41.53','A41.59','A41.81','A41.89', 'A41.9'
- **SEVERE SEPSIS:** 'R65.20','R65.21','R65.10','R65.11'

Methods: processing of clinical notes: Please cite the implementation (software package) that was used, as well as for your prediction model.

The text mining procedure were conducted using SAS Enterprise Miner 14.1 and the Ensemble machine learning was conducted using KNIME Analytics Platform (version 4.1.6).

The paper is missing a demographics table to describe the patient population. For example, what is the incidence of sepsis, severe sepsis, septic shock? How many are admitted to the ICU? What is the age distribution?

	Training/validation	Testing
Number of patients	3722	1595
Age - years old	63.71 ± 17.08 (Mean ± SD)	63.90±16.81 (Mean±SD)
Male - %	57.3	60.67
Length of hospital stay - days	5.52 ± 14.31 (Mean ± SD)	5.17±10.80 (Mean±SD)
ICU Admission - %	7.52	8.61
Mortality - %	4.5	5.01
Septic - %	6.45%	5.45%
Non-septic - %	93.55%	94.55%

What was the class imbalance of your prediction and how did you account for it? It is unclear from the manuscript if a balanced dataset was created by randomly under sampling the non-sepsis cohort, or if the class imbalance was dealt with in some other manner during training.

The original dataset is imbalanced as it consists of data extracted from a single hospital over a 2.5 year period. We sampled the data so that the prevalence of the cases would be similar to natural prevalence in clinical settings. We selected **all** sepsis patient (cases) and randomly selected non-sepsis cases (controls) and arrived at **patient-visit level** prevalence of **6.15%**. The level of prevalence is equivalent to the natural prevalence of sepsis typically observed in hospitals. As seen in (Rhee et al., 2017) from 2009 to 2014 the prevalence of sepsis is about 6% of the patient population and it relatively stable over time. (cf. pp. 1246 (Rhee et al., 2017)).

In our initial submission, we trained/validated and tested the model **without** any oversampling procedure applied to the data. Due to the low prevalence of sepsis in our dataset, and given that the analysis was done using each **clinical note** as the **unit of analysis**, the prevalence of sepsis in the clinical was around 1% for the early prediction algorithm leading to a naturally low PPV, even with AUC of > 0.90 and sensitivity 0.86 and specificity 0.80.

Based on the review team’s suggestion to test our model under higher prevalence of sepsis, as seen in most machine learning studies where some form of over/under sampling is used (Liu et al., 2019), we oversampled the sepsis cases using SMOTE (Synthetic Minority Oversampling Technique) for the training and validation dataset. SMOTE is a commonly applied oversampling procedure where additional positive cases are imputed via a nearest neighbor resampling algorithm (Chawla et al., 2002). This method is used in prior studies published in *Nature Communications* that develop machine learning classifiers in low prevalence environment, e.g., oral cancer detection(Carnielli et al., 2018) and cell identification/ classification(Rennie et al., 2018; Xia et al., 2020).

It is important to note that the AUC, sensitivity, and specificity of models developed using oversampling (SMOTE) and models developed without oversampling are very similar as seen in Tables 3a to 3f (see response to Reviewer 1). The PPV for models without oversampling are significantly lower as PPV is algebraically constrained by the prevalence of sepsis (see equation 2 below) (only exception is where specificity equals to 1). With the low prevalence in our sample, we expected low PPV values.

$$PPV = \frac{\text{sensitivity} \times \text{prevalence}}{(\text{sensitivity} \times \text{prevalence}) + [(1 - \text{specificity}) \times (1 - \text{prevalence})]} \quad \text{Eq. 2}$$

It would be helpful if a standard CONSORT enrolment diagram was included as a figure, potentially to replace Figure 1.

Thank you for this useful suggestion and we agree that the current Figure 1 is less informative. However, your suggestion of a CONSORT diagram is applicable only to a randomized controlled trial but ours is a case-control study. As such, we have used a STROBE Enrolment diagram to replace Figure 1 as suggested in Vandembroucke et al. (2007). We believe a STROBE Enrolment diagram will provide the equivalent information as a CONSORT diagram.

It would also be helpful if one included a reliability diagram (calibration diagram) as well as a precision-recall diagram for a representative model to better understand calibration as well as the trade-offs between precision and recall for choosing a decision threshold.

Thank you for raising this point. We have now included the calibration curves (Figure R1 in response to Reviewer 1 comments) as well as the precision-recall curves below (Figures R4).

* Note that the Precision axis is truncated and starts at 0.5.

Figure R4: Precision – Recall Curve (SERA algorithm)

REFERENCES

- Batista, G. E., Prati, R. C., & Monard, M. C. (2004). A study of the behavior of several methods for balancing machine learning training data. *ACM SIGKDD explorations newsletter*, 6(1), 20-29.
- Blei, D. M. (2012). Probabilistic topic models. *Communications of the ACM*, 55(4), 77-84.
- Blei, D. M., Ng, A. Y., & Jordan, M. I. (2003). Latent dirichlet allocation. *Journal of Machine Learning Research*, 3(Jan), 993-1022.
- Carnielli, C. M., Macedo, C. C. S., De Rossi, T., Granato, D. C., Rivera, C., Domingues, R. R., . . . Paes Leme, A. F. (2018). Combining discovery and targeted proteomics reveals a prognostic signature in oral cancer. *Nature communications*, 9(1), 3598. doi:10.1038/s41467-018-05696-2
- Chawla, N. V., Bowyer, K. W., Hall, L. O., & Kegelmeyer, W. P. (2002). SMOTE: synthetic minority over-sampling technique. *Journal of artificial intelligence research*, 16, 321-357.
- Horng, S., Sontag, D. A., Halpern, Y., Jernite, Y., Shapiro, N. I., & Nathanson, L. A. (2017). Creating an automated trigger for sepsis clinical decision support at emergency department triage using machine learning. *PLoS one*, 12(4).
- Kumar, A., Roberts, D., Wood, K. E., Light, B., Parrillo, J. E., Sharma, S., . . . Taiberg, L. (2006). Duration of hypotension before initiation of effective antimicrobial therapy is the critical determinant of survival in human septic shock. *Critical Care Medicine*, 34(6), 1589-1596.
- Liu, R., Greenstein, J. L., Sarma, S. V., & Winslow, R. L. (2019). *Natural Language Processing of Clinical Notes for Improved Early Prediction of Septic Shock in the ICU*. Paper presented at the 2019 41st Annual International Conference of the IEEE Engineering in Medicine and Biology Society (EMBC).
- Rennie, S., Dalby, M., van Duin, L., & Andersson, R. (2018). Transcriptional decomposition reveals active chromatin architectures and cell specific regulatory interactions. *Nature communications*, 9(1), 487. doi:10.1038/s41467-017-02798-1
- Rhee, C., Dantes, R., Epstein, L., Murphy, D. J., Seymour, C. W., Iwashyna, T. J., . . . Fiore, A. E. (2017). Incidence and trends of sepsis in US hospitals using clinical vs claims data, 2009-2014. *Jama*, 318(13), 1241-1249.
- Rhodes, A., Evans, L. E., Alhazzani, W., Levy, M. M., Antonelli, M., Ferrer, R., . . . Nunnally, M. E. (2017). Surviving sepsis campaign: international guidelines for management of sepsis and septic shock: 2016. *Intensive care medicine*, 43(3), 304-377.
- Vandembroucke, J. P., Von Elm, E., Altman, D. G., Gøtzsche, P. C., Mulrow, C. D., Pocock, S. J., . . . Initiative, S. (2007). Strengthening the Reporting of Observational Studies in Epidemiology (STROBE): explanation and elaboration. *PLoS Med*, 4(10), e297.
- Wallach, H. M. (2006). *Topic modeling: beyond bag-of-words*. Paper presented at the Proceedings of the 23rd international conference on Machine learning.

- Xia, B., Zhao, D., Wang, G., Zhang, M., Lv, J., Tomoiaga, A. S., . . . Chen, K. (2020). Machine learning uncovers cell identity regulator by histone code. *Nature communications*, 11(1), 2696. doi:10.1038/s41467-020-16539-4

Reviewers' Comments:

Reviewer #1:

Remarks to the Author:

In general the authors have responded to the majority of my concerns.

I feel my final comment on the practicability of SERA algorithm in clinical practice could have been addressed better. As shown in [1] traditional machine learning methods dominate clinical practice with respect to deep learning methods. As such, the authors could use this evidence to better motivate their discussion.

[1] Sheikhalishahi et al., "Natural Language Processing of Clinical Notes on Chronic Diseases: Systematic Review" JMIR Med Inform, DOI:10.2196/12239, 2019

REVIEWERS' COMMENTS

Reviewer #1 (Remarks to the Author):

In general, the authors have responded to the majority of my concerns. I feel my final comment on the practicability of SERA algorithm in clinical practice could have been addressed better. As shown in [1] traditional machine learning methods dominate clinical practice with respect to deep learning methods. As such, the authors could use this evidence to better motivate their discussion.

[1] Sheikhalishahi et al., "Natural Language Processing of Clinical Notes on Chronic Diseases: Systematic Review" JMIR Med Inform, DOI:10.2196/12239, 2019

Thank you for your comments. In this revision, we have revised the discussion section using Sheikhalishahi et al. as the basis for motivating the use of NLP for clinical applications.